# Multimodal Imaging of Microvascular Abnormalities in Retinal Vein Occlusion

**DOI:** 10.3390/jcm10030405

**Published:** 2021-01-21

**Authors:** Yoshio Hirano, Norihiro Suzuki, Taneto Tomiyasu, Ryo Kurobe, Yusuke Yasuda, Yuya Esaki, Tsutomu Yasukawa, Munenori Yoshida, Yuichiro Ogura

**Affiliations:** Department of Ophthalmology and Visual Science, Nagoya City University Graduate School of Medical Sciences, Nagoya 4678601, Japan; n.suzuki11@gmail.com (N.S.); tanetomas@yahoo.co.jp (T.T.); monokurokuroboo1025@gmail.com (R.K.); yy4amm1111@gmail.com (Y.Y.); smart.uribou.529@gmail.com (Y.E.); yasukawa@med.nagoya-cu.ac.jp (T.Y.); muney@med.nagoya-cu.ac.jp (M.Y.); ogura@med.nagoya-cu.ac.jp (Y.O.)

**Keywords:** retinal vein occlusion, macular edema, microaneurysm, multimodal imaging, vascular endothelial growth factor, pathology

## Abstract

The technologies of ocular imaging modalities such as optical coherence tomography (OCT) and OCT angiography (OCTA) have progressed remarkably. Of these in vivo imaging modalities, recently advanced OCT technology provides high-resolution images, e.g., histologic imaging, enabling anatomical analysis of each retinal layer, including the photoreceptor layers. Recently developed OCTA also visualizes the vascular networks three-dimensionally, which provides better understanding of the retinal deep capillary layer. In addition, ex vivo analysis using autologous aqueous or vitreous humor shows that inflammatory cytokine levels including vascular endothelial growth factor (VEGF) are elevated and correlated with the severity of macular edema (ME) in eyes with retinal vein occlusion (RVO). Furthermore, a combination of multiple modalities enables deeper understanding of the pathology. Regarding therapy, intravitreal injection of anti-VEGF drugs provides rapid resolution of ME and much better visual improvements than conventional treatments in eyes with RVO. Thus, the technologies of examination and treatment for managing eyes with RVO have progressed rapidly. In this paper, we review the multimodal imaging and therapeutic strategies for eyes with RVO with the hope that it provides better understanding of the pathology and leads to the development of new therapies.

## 1. Introduction

Retinal vein occlusion (RVO) is the second most common retinal vascular disorder next to diabetic retinopathy. In most patients with RVO, macular edema (ME) is the pre-dominant cause of visual loss in the acute and chronic stages. Intravitreal (IV) injection of anti-vascular endothelial growth factor (VEGF) agents has revolutionized the therapy for ME associated with RVO [1,2]. However, the frequent recurrence of ME after anti-VEGF therapies highlight the need for better understanding of the pathology of ME and development of a therapy based on the pathology. Various microvascular abnormalities (Figure 1) are observed in eyes with RVO, which complicate the pathology in eyes with RVO.

To date, various advanced diagnostic tools for functional and structural analyses have been developed. Using multiple modalities such as fundus photography, optical coherence tomography (OCT), OCT angiography (OCTA), fluorescein angiography (FA), indocyanine green angiography (ICGA), and ultrawide field angiography, we review the multimodal imaging of microvascular abnormalities and therapeutic strategies in eyes with RVO.

## 2. Mechanism of Retinal Vein Occlusion

### 2.1. Pathogenesis of Retinal Vein Occlusion

The pathogenesis of RVO is multifactorial. Arteriosclerosis strongly contributes to the onset of RVO. Arteriosclerosis results in vein occlusion through endothelial cell damage and thrombosis. For instance, a rigid artery compresses the underlying venous wall at the arteriovenous (AV) crossings, resulting in disturbance of venous return and changes in the course of venous flow, turbulent blood flow, chronic damage to the retinal vascular endothelial cells, and thrombosis formation, which result in onset of RVO. A thrombosis is thought to be caused by the three important factors of Virchow’s triad, i.e., endothelial injury, hypercoagulability, and abnormal blood flow (Figure 2). It is reasonable that these three factors strongly contribute to the pathology of RVO. 

Increased vascular pressure behind the occlusion may lead to leakage of fluid across the vascular wall to the adjacent retinal tissue [3,4,5,6]. Furthermore, the damage of the endothelium in the affected vein may induce a low-grade, chronic inflammation of the retinal microvasculature and an upregulation of inflammatory mediators that break the blood–retina barrier and perpetuate ME [3,4,5,6]. 

Another hypothesis is that arteriosclerosis results in arterial insufficiency, leading to RVO. Arterial insufficiency, i.e., insufficient oxygen transport to the retina due to arteriosclerosis, can cause retinal hypoxia, resulting in production of VEGF, which plays a central role in the onset of and various microvascular abnormalities in RVO.

Multiple studies have established a significance of AV crossings in eyes with branch retinal vein occlusion (BRVO) [7,8,9]. Namely, the artery and vein share the common adventitial sheath at the AV crossings and the rigid artery causes a mechanical obstruction of the vein (Figure 2). Further, thickened vessel walls and narrowed venous lumina also result in RVO, as a histological study reported that used postmortem tissue from a patient with central RVO (CRVO) [10]. Previous studies using OCT also have shown that the retinal veins narrowed at the AV crossings [11]. The paper described that in all eyes with BRVO of the vein overcrossing, the vein appeared to be compressed and choked between the internal limiting membrane and the arterial wall at the AV crossing, whereas the venous lumen was generally preserved at the AV crossing in eyes with arterial overcrossing [11], suggesting that other factors such as shear stress due to abnormal blood flow and/or endothelial injury might cause the pathogenesis of BRVO (Figure 2).

### 2.2. Endothelial Injury and Thickening of the Vessel Walls

Vascular endothelial cells maintain smooth microcirculation and play a central role in abnormal microcirculation (Figure 2). Vascular endothelial cells usually regulate vascular resistance, i.e., prevent thrombosis formation by producing vasodilator and vasoconstrictor mediators such as nitric oxide (NO), prostaglandin I2 (PGI2), and endothelin. However, chronic stress to the endothelium due to hypertension, diabetes mellitus, and hyperlipidemia causes arteriosclerosis. A mechanical obstruction of the vein at the AV crossings results in venous blood stasis, turbulent blood flow, endothelial injury, thickening of the intima media, and formation of a vascular thrombosis. A histologic study using postmortem tissue from a patient with CRVO [10] reported thickened vessel walls and narrowed venous lumens. Furthermore, previous studies using OCT have also shown that the venous lumen narrowed at the AV crossings in eyes with BRVO [11]. Loss of capillary endothelial cells and pericytes has been reported in previous histological studies [10,12]. Moreover, retinal vascular endothelial cells form a tight junction with each other, function as an inner blood–retina barrier, and maintain the retinal stability. Therefore, the retinal vascular endothelial injury and consequent blood–retina barrier breakdown cause vascular hyperpermeability and the retina becomes edematous. VEGF also has been reported to increase vessel permeability by increasing the phosphorylation of tight junction proteins and was thus an important mediator of the blood–retina barrier breakdown leading to vascular leakage and ME [13]. We can indirectly evaluate the endothelial injury using FA, which can image the hyperpermeability (Figure 3).

### 2.3. Hypercoagulability

Vascular endothelial cells control platelet function and blood coagulation and fibrinolysis. NO and PGI2 are produced by endothelial cells and not only cause vasodilation but also impede platelet aggregation and control the microvascular circulation by interacting with each other. Therefore, venous blood stasis at the AV crossings damages the retinal vascular endothelial cells, leading to decreased production of NO and PGI2 and increased production of tissue factors, resulting in hypercoagulability and acceleration of thrombosis formation (Figure 2).

### 2.4. Abnormal Blood Flow

Yoshida et al. [14], who measured retinal blood vessel diameters, retinal blood flow, and the absolute retinal blood velocity of the 18 arterial sites and 18 venous sites between the optic disc margin and the first bifurcation in healthy subjects and a patient with RVO using a laser Doppler flowmeter, reported that the blood velocities in the eye with BRVO and the fellow eye decreased compared with that in a normal eye. Moreover, the flow characteristics in a patient with CRVO improved dramatically after the occlusion resolved. Yamada et al. [15] also reported that the mean blur rate of the major vessels around the optic disc in eyes with CRVO measured by laser speckle flowgraphy decreased compared with that in the fellow eye and was correlated inversely with the aqueous VEGF concentration level. The oxygen saturation in first- and second-degree retinal venules also decreased in the patient with CRVO [16]. Since the bloodstream transports both nourishment and oxygen, decreased retinal blood flow can cause hypoxia in the retina, leading to production of hypoxia-inducible factors and consequent VEGF expression (Figure 2).

Blood flow also is associated with shear stress on the vessel walls, i.e., appropriate shear stress on the vessel walls causes endothelial cells to secret NO, which prevents thrombosis formation by controlling platelet aggregation and leukocyte adhesion to the vessel walls. In contrast, decreased shear stress on the vessel walls results in decreased inflammatory cytokines and cell adhesion molecules and the increased possibility of leukocyte adhesion and thrombosis formation (Figure 2).

## 3. Multimodal Imaging of Microvascular Abnormalities in Retinal Vein Occlusion

### 3.1. Thrombosis

Using OCT, thrombi were detected around the AV crossing in eyes with RVO [11]. Interestingly, the thrombi were always downstream of the AV crossing, and no eye had thrombi only at the crossing site or upstream of the AV crossing alone [11]. Histological studies [8,12] have also demonstrated thrombus formation in eyes with BRVO. In contrast, Seitz reported [17] that no blood thrombi obliterated the venous lumens at the AV crossings, suggesting that venous compression is one of the important factors in the pathogenesis of BRVO. Thus, it is unclear whether thrombus formation is a cause or result of BRVO.

### 3.2. Retinal and Subretinal Hemorrhage

Both hemorrhagic types are observed in acute RVO and can cause acute visual loss, the effect of which is inferior to ME. Subretinal hemorrhages especially can damage the foveal photoreceptors, causing consequent visual loss [18]. IV injection of anti-VEGF agents accelerated absorption of the subretinal hemorrhage, thus preventing damage to the foveal photoreceptors and consequent visual preservation [19].

### 3.3. Macular Edema

ME and retinal and subretinal hemorrhages occur at the acute stage in eyes with RVO, both of which can cause acute visual loss. Although ME is the main cause of visual loss in eyes with RVO at the acute and chronic stages, macular ischemia, atrophy, and hemorrhages can be confounders that reduce the correlation of ME with the visual outcomes in eyes with BRVO and hemi-CRVO [20] (Figure 4).

ME in eyes with RVO develops as the result of elevated venous pressure and injury to the retinal vascular endothelial cells, which act as an inner blood–retina barrier. In eyes with BRVO, intraocular levels of VEGF, interleukin (IL)-6, soluble VEGF receptor (VEGFR)-2, and soluble intercellular adhesion molecule 1 (ICAM-1) are elevated and correlated with the severity of ME [13,21,22,23]. Of these cytokines, VEGF plays an important role in the formation of ME, i.e., VEGF activates VEGFR-2, which causes the decreased expression of occluding in tight junctions and breakdown of the blood–retina barrier, leading to leakage of proteins, lipids, and blood into the retina, which is ME [23,24]. VEGF also stimulates endothelial production of NO, which is a vasodilator. VEGF also increases the expression of ICAM-1.

Thus, anti-VEGF therapy theoretically reduces NO production, leading to vasoconstriction. Vasoconstriction reduces ME independent of the effect on vascular permeability [25]. However, the ME in RVO is not entirely always mediated by VEGF, because in some cases, blockade of VEGF by IV injection of anti-VEGF agents fails to resolve the ME [1].

The detection modalities for ME vary, e.g., slit-lamp biomicroscopy, OCT, and FA, of which OCT is the most sensitive and the most commonly used to evaluate ME. The central subfield thickness (CST), defined as the mean thickness of the central 1-mm circle centered on the fovea, is the most used parameter to evaluate ME. CST parameters are used to evaluate the disease activity in standard clinical care and as the inclusion and retreatment criteria and the morphologic endpoint in randomized clinical trials [1,26].

OCT usually shows a combination of cystoid ME, sponge-like retinal swelling, and serous retinal detachment (RD) [27]. In particular, tiny serous RD is only detectable on OCT images (Figure 3). Besides, recently advanced OCT technology provides high-resolution images, such as histologic imaging that enables anatomic analysis of each retinal layer. Consequently, several morphologic features such as intraretinal fluid, subretinal fluid, and hyperreflective foci, and differential retinal layer analysis, indicating whether these layers are thickened, thinned, or disrupted or not in the photoreceptor layers, can be distinguished [28].

Because the resolution of OCT images has improved markedly, new findings such as the foveal bulge [29], track line [30], disorganization of the inner retinal layers [31], prominent middle limiting membrane [32], paracentral acute middle maculopathy [33], integrity of external limiting membrane [34,35], or the inner and outer photoreceptor segment lines [36] have been examined by OCT. The relationships of these factors with visual function also have been studied in eyes with ME [29,30,31,32,33,35,36].

The status of vitreous or vitreoretinal interface is also associated with ME in eyes with RVO. High-resolution OCT, especially swept-source OCT, visualizes the status or vitreous or vitreomacular interface [37]. Cases with a non-posterior vitreous detachment (PVD) or partial PVD had a significantly higher incidence of development of ME than cases with a total PVD [38]. Furthermore, non-PVD cases with nonischemic CRVO had a significantly higher incidence of development of ME than cases with a total PVD [39]. For ischemic CRVO, however, there was no interaction between the vitreous status and ME. ME could be present in eyes with ischemic CRVO even when the posterior vitreous was detached [39].

The central macula including the foveal avascular zone (FAZ) is generally the thickest, probably because the center of the macula is avascular, and therefore, the only mechanism for drainage of the intraretinal fluid is the retinal pigment epithelium pump.

Müller cells are important for transporting water from the extracellular space into the retinal capillaries of the inner retina [40]. Therefore, Müller cells’ dysfunction contributes to ME. The Müller cell cone, initially described by Yamada [41], may serve as a plug that binds the photoreceptor cells in the foveola [42,43], the latter of whom speculated that the Müller cell cone plays an important role in the occurrence serous RDs in eyes with RVO.

The distribution of ME in eyes with RVO differs between the acute and chronic stages. Chronic ME in eyes with RVO is often localized to the macular center, while acute ME is not. Localized chronic macular edema often is caused by microaneurysms [44].

In eyes with chronic ME, the thickness of the foveal photoreceptor layer was correlated with visual acuity (VA), but the foveal thickness was not [45]. Therefore, chronic ME with intact foveal photoreceptor layers may be observed without intervention if the VA has not deteriorated (Figure 5).

Although advanced OCT technology provides detailed morphologic features, only cross-sections across the fovea are usually examined. OCT color maps can identify the parafoveal or extrafoveal edema and the central area. Moreover, a multimodal imaging technique enables detection of the leakage points causing ME (Figure 6 and Figure 7). Specifically, an OCT color map provides the shape and range of ME-like topography (Figure 6 and Figure 7). Since the leakage points causing ME should be around the center of the edema, the tops of the points of edema appear on the OCT color map. Further, superimposed images of the OCT color map and FA and/or ICGA (Figure 6) may identify the source of the ME, leading to successful application of the laser treatment. Although OCTA enables differential retinal layer analysis, it is not effective in eyes with ME due to segmentation errors.

The relations between macular ischemia and ME and visual outcomes have been investigated. A disrupted FAZ border is correlated strongly with macular ischemia [46]. OCTA can clearly visualize the disrupted FAZ border (Figure 8). A higher percentage of eyes with ischemic ME associated with BRVO have spontaneous resolution of ME than eyes with nonischemic ME [46]. Furthermore, patients with a large reduction in the number of macular vessels in OCTA findings have fewer recurrences and a lower frequency of IV ranibizumab injections [47]. Regarding the visual outcomes of eyes with ischemic ME in RVO, Finkelstein [46] reported favorable results and others reported unfavorable results [48,49].

Several previous studies have investigated the relation of collateral vessel formation and resolution of ME and subsequent visual outcomes. Collateral vessel formation decreases the retinal venous pressure and vascular permeability, leading to spontaneous resolution of ME [9]. Although Priluck et al. [50] reported that the presence of collateral vessels was associated with visual improvement in eyes with CRVO, most previous reports found no association between collateral vessel formation and visual improvements [51,52,53,54,55].

### 3.4. Retinal Nonperfusion

Nonperfused areas (NPAs) form in association with retinal artery or vein occlusion. The mechanism of NPA formation remains unclear. Compression of capillaries by swollen retinal tissues [56], arterial insufficiency based on back pressure by the obstructed veins [57,58], or capillary occlusion due to leukocyte accumulation at the endothelium [59] has been speculated as causative.

Nonperfusion is often considered similar to ischemia; however, they are distinct entities [60], i.e., there is some degree of perfusion even if the retina is ischemic. It takes about 3-4 weeks of ischemia before the retinal capillaries become nonperfused [60]. In a murine model, NPAs formed about 7 days after laser-induced CRVO [61]. The size of the NPAs can help distinguish between the ischemic and nonischemic types in eyes with RVO. NPAs are supposed to secrete VEGF due to lack of blood flow and consequent insufficient oxygen supply. The VEGF level in the aqueous humor was reported to be higher in ischemic RVO than in nonischemic RVO [62]. VEGF overexpression promotes leukocyte accumulation, possibly triggering aggravation of the capillary occlusion, which is why enlargement of NPAs is seen frequently during follow-up in eyes with RVO.

OCTA is superior to FA for detecting NPAs in eyes with RVO because of the longer wavelength, no use of fluorescein, and higher-resolution images [63] (Figure 9). OCTA also showed that NPAs are larger in the retinal deep capillary plexus than in the retinal superficial capillary plexus [63], possibly because the capillary density in the deep plexus is higher than in the superficial plexus [64]. Interestingly, Iida et al. [65] reported that OCTA showed a significant association between NPA size and AV crossing pattern in eyes with BRVO, i.e., the NPAs were larger in eyes with venous overcrossing than in those with arterial overcrossing. Muraoka et al. [11] showed that this occurred because occluded veins in the venous overcrossing type were compressed between the internal limiting membrane (ILM) and the arterial wall.

Finkelstein [46] found that ischemic edema is often transient and associated with good visual outcomes. In addition, macular vessel reduction measured by OCTA and patients with a larger reduction in macular vessels have fewer recurrences of ME [47], i.e., severe ischemia may not produce VEGF and consequent recurrent ME.

Retinal sensitivities measured by microperimetry in the NPAs were investigated in eyes with RVO and deteriorated in the NPAs compared with perfused areas [66,67] (Figure 10). We found that the retinal sensitivities in NPAs were correlated inversely with the duration after disease onset [67] (Figure 10), and ME and microaneurysms were unlikely to be around the area of decreased retinal sensitivity (Figure 11), i.e., severe ischemia where the retina already did not function might not produce VEGF proteins. In addition, the retinal sensitivity at the border between NPAs and perfused areas was relatively preserved and ME and microaneurysms were more frequently observed there [67], which is consistent with the fact that the borders are the common sites of microaneurysms and retinal neovascularization (NV) formation.

### 3.5. Foveal Avascular Zone: The Enlargement and the Circularity

The FAZ is in the macular center, the size is about 400 μm in diameter, and retinal blood vessels are not seen in the area. Therefore, the oxygen supply in the area depends on the choroidal circulation.

The FAZ in eyes with BRVO is enlarged compared with that in control subjects [68]; however, this has not received a great deal of attention. OCTA can visualize the FAZ better than FA and makes it possible to investigate the differential retinal layer analysis (Figure 9) [69]. The FAZs in the retinal superficial and deep capillary plexuses detected by OCTA are enlarged in eyes with RVO [69,70]. Moreover, the FAZs in both layers enlarged gradually during follow-up [69]. It was also reported that the size of the FAZ in the retinal deep capillary plexus is correlated with the VA [70,71,72]. Seknazi et al. [73] reported that the size of the FAZ and macular vascular density were correlated positively with the size of the peripheral NPAs and speculated that reduced retinal vessel density may predict development of retinal NV in eyes with RVO. The foveal capillary ring, which borders the FAZ, is better visualized by OCTA than FA (Figure 9), is occasionally disrupted, and consequently, the border between the FAZ and neighboring NPAs becomes unclear in eyes with RVO, which might be an emerging issue because OCTA can clearly depict the NPAs and FAZ. Another study found that disruption of the foveal capillary ring is correlated with peripheral retinal ischemia in eyes with RVO [74]. Further, the acircularity index of the FAZ is defined as the ratio of the perimeter of the FAZ and the perimeter of a circle with equal area [75], and a perfectly circular FAZ has an acircularity equal to 1 (Figure 12). Deviations from a circular shape increase the value of this acircularity metric [76]. The acircularity index of FAZ was investigated as a biomarker of visual outcomes, although it was found to not be one [76].

Although OCTA enables to visualize and measure the FAZ in a non-invasive manner, the FAZ can vary in size and shape even in normal subjects [77,78,79,80]. Therefore, measurements of FAZ obtained using OCTA should be analyzed carefully.

### 3.6. Dilated Tortuous Veins and Capillary Telangiectasia

Capillary telangiectasia associated with elevated venous pressure due to RVO, which occurs frequently in eyes with RVO. OCTA visualized capillary telangiectasia in the retinal superficial and deep capillary plexuses of eyes with RVO, and capillary telangiectasia was more common and more widespread in the deep capillary plexus than in the superficial capillary plexus (Figure 9) [63,74]. For instance, our previous study using OCTA [63] reported that 13 eyes (46.4%) and 28 eyes (100%) of 28 eyes had capillary telangiectasia in the superficial and deep capillary plexuses, respectively, possibly because the capillary density in the deep plexus is higher than in the superficial plexus [64]. Thus, the ability to provide information about the deep capillary plexus in vivo would be an advantage of OCTA because FA does not visualize the retinal deeper capillary network well, possibly because of light scattering in the retina [63,72,81]. Nowadays, OCTA can visualize four morphologically different capillary networks including a radial peripapillary capillary plexus, which is only around the optic disc, and superficial, intermediate, and deep capillary plexuses [82] (Figure 13). Immunohistochemical analysis using postmortem eye tissue from a patient with CRVO also showed that the veins in the affected area were dilated and tortuous. Furthermore, the vessel walls were thicker than those in the non-affected area, the luminal diameters decreased, and the endothelial cells and mural cells died in the affected area [10].

### 3.7. Collateral Vessels

Collateral vessels are defined as dilated vessels or capillary telangiectasia arising between the affected and healthy venules [83]. Collateral vessels develop from the preexisting retinal capillary network rather than the formation of new vessels to drain blood flow from an obstructed vein into an adjacent area [9,84,85] and are distinguished from retinal NVs resulting from a microvascular response to VEGF [86,87]. FA showed that collateral vessels usually do not leak in contrast to retinal NV [85]. Although the difference on FA facilitates distinguishing these abnormalities, OCTA can visualize collateral vessels better than FA, because OCTA provides higher-resolution images than FA without fluorescein leakage [55,63].

Collateral vessels were best developed around the FAZ and over the temporal watershed zone (Figure 14) [9,55,83]. In eyes with BRVO, most collateral vessels are within the retina and rarely on the optic disc. In contrast, in eyes with hemi-CRVO or CRVO, col-lateral vessels are occasionally seen at the optic disc. The differences in the features between disc collaterals and NV of the disc (NVD) on OCTA were as follows: collateral vessels on the optic disc appeared as small, loopy vessels distinct from surrounding peripapillary capillaries on OCTA in the radial peripapillary capillary frame, and retinal NVD appeared as a mesh of fine-caliber raised vessels best seen in the vitreous slab of OCTA [88].

Regarding the origin of the collateral vessels’ development, collateral vessel formation has been reported to be driven by hydrodynamic and hydrostatic forces that cause existing capillary channels to enlarge to bypass venous obstruction [84,85,89]. The pressure gradient between an obstructed vein and neighboring unobstructed vessels is sufficient to explain the blood flow through the collateral channels in RVOs [89]. We previously reported that collateral vessels were seen more frequently in eyes with major BRVO or the ischemic type than those with macular BRVO or the nonischemic type [55]. That study also showed that eyes with collateral vessels have a greater baseline central retinal thickness (526 ± 151 μm) and a higher proportion of major BRVO (18 of 23 eyes, 78.3%) than eyes without collateral vessels (362 ± 71 μm and 0 of 5 eyes, 0%) [53], suggesting that a large amount of congested venous blood may be related to collateral vessel development. The association of retinal ischemia with collateral vessel formation has been supported by other studies [83,90,91]. The larger area of capillary nonperfusion might be a marker for greater hemodynamic stress and a larger pressure gradient in the retinal venous system, facilitating intraretinal collateral formation [90]. Tsuboi et al. [83] speculated that the presence of widespread NPAs forces the residual capillaries to enlarge to bypass venous obstruction, resulting in collateral formation, and proposed that the dilated capillaries represent a kind of vessel remodeling. The presence of collateral vessels at the optic disc in eyes with CRVO was a negative predictor of development of anterior segment NV, suggesting that optic disc collaterals may protect against ischemia [92].

Three types of collaterals have been described, i.e., the arterioarteriolar (AA) pattern that develops after branch artery obstruction; the venovenular (VV) pattern that develops after vein obstruction; and the AV pattern that occurs in capillary bed obstruction [85].

Freund et al. [93] reported that collateral vessels in eyes with RVO were exclusively in the deep capillary plexus. FA usually visualizes collateral vessels. However, FA does not image the deep capillary networks well [94]. Other reports have shown that collateral vessels are in the superficial and deep capillary plexuses [55,65,83]. In addition, Lee et al. [91] classified collateral vessels into four types: true superficial vessels that course exclusively through the superficial capillary plexus; true deep vessels that course exclusively through the deep capillary plexus; superficial diving vessels that connect two superficial vessels via the deep capillary plexus; and foveal collaterals that cross the horizontal raphe across the fovea and connect two superficial blood vessels (Figure 14). Forty-three collateral vessels then were identified: 19 true superficial, 9 true deep, 10 superficial diving, and 5 foveal collaterals—12 of the 19 (63%) true superficial collaterals developed in eyes with CRVO, while all 10 superficial dividing collaterals (100%) developed in eyes with BRVO [91]. Moreover, the authors found that true superficial collaterals were all AV patterns, while true deep collaterals and diving collaterals were all VV patterns [91].

Several studies have reported the treatment effects on collateral vessel formation. In the human study, IV corticosteroids neither enhanced nor inhibited the development of collaterals in eyes with RVO [90]. IV injection of anti-VEGF agents (ranibizumab, Lucentis, Genentech, Inc., South San Francisco, CA, USA) did not affect the incidence of collateral vessel formation on the disc or in the retina in patients in the BRAVO or Ranibizumab for the Treatment of Macular Edema after Central Retinal Vein OcclUsIon Study: Evaluation of Efficacy and Safety (CRUISE) studies [95].

Collateral vessels in eyes with RVO are considered to be chronic. Collateral vessels mature over 6 to 24 months after BRVO onset [9]. In contrast, previous animal studies have shown that collateral vessels were observed within 3 to 5 days after BRVO [84,96,97]. One study [97] found that collateral vessels developed from the preexisting capillary networks up to 4 days after laser-induced vein occlusion, matured by day 7, and were pruned on day 14 to a size similar to the original retinal vein. Thus, the duration of collateral formation after experimental RVO in animals is much shorter than in human eyes because the animals in the experiments were young and healthy, i.e., they differ from older patients with systemic diseases such as diabetes mellitus, hypertension, and hyperlipidemia [98]. Collaterals also were reported in the retina at baseline in 65 (17%) of 381 BRVO eyes [90], but the grading of collaterals was performed using only color fundus photographs, not FAs, indicating a possible inaccuracy. We previously reported that OCTA can detect collaterals better than FA [55,63]. Furthermore, all collateral vessels detected by OCTA in eyes with BRVO were already formed at the initial visit (Figure 14) [55]. The pressure gradient between an obstructed vein and neighboring unobstructed vessels is sufficient to explain the blood flow through the collateral channels in RVOs [89]. Therefore, it is reasonable that collateral vessels exist at baseline because the venous pressure is usually elevated after RVO onset.

Controversy exists over whether collateral vessel formation is beneficial to the resolution of ME. Once collateral vessels form, the retinal venous pressure and vascular permeability decrease and lead to spontaneous resolution of ME [5]. In contrast, Hayreh et al. [52] reported that collateral vessel development did not promote the resolution of ME in eyes with CRVO. We previously reported that eyes with collateral vessels had greater reductions in central retinal thickness [55]. However, eyes with collateral vessels have a greater baseline central retinal thickness than eyes without collateral vessels, which may explain why the central retinal thickness reduction rate was greater in eyes with collateral vessels. Besides, we reported that collateral formation was associated with refractory ME in eyes with BRVO [38], possibly because of leaky microaneurysms formed in the collaterals (Figure 15) [55,63]. Moreover, the number of collateral vessels around the fovea was correlated with the number of IV injections of anti-VEGF agents, and the collateral vessels around the fovea may be a good indicator of persistent ME [84]. Thus, collateral vessel formation may negatively affect the resolution of ME. Several reports [52,55,90,95] have reported that the effect of collateral vessel formation on the VA seems not to be beneficial.

### 3.8. Microaneurysm and Macroaneurysm

A microaneurysm is defined as a red structure smaller than 100 micrometers and a macroaneurysm as a structure 100 micrometers or larger [99]. Both occur in systemic and retinal diseases including ischemic, infectious, inflammatory, and hematologic disorders [56,100,101,102,103,104], can be important signs of progression of systemic disease [104,105], and are components of multiple retinopathy severity classification scores [104,106].

Pericytes and endothelial cells are key components of the neurovascular unit (Figure 16). A histopathologic study revealed that pericytes and endothelial cells were rarely present in the area of RVO [12]. These conditions weaken the capillary wall and lead to the formation of microaneurysms and macroaneurysms [107]. Furthermore, elevated intraocular VEGF levels have been reported in eyes with RVOs [13,108]. Together, vascular endothelial cells cause proliferative changes.

An animal study found that IV injection of VEGF proteins produced retinal ischemia and microangiopathy such as microaneurysms in adult monkey eyes [59]. We previously reported in a retrospective study that anti-VEGF therapy suppressed microaneurysm formation compared with steroid therapy (either sub-Tenon capsule (ST) injection or IV injection of triamcinolone acetonide (TA)) [44] (Figure 17). In addition, prompt administration (within 3 months after disease onset) of anti-VEGF agents was more effective than delayed administration (over 3 months after disease onset) [44], which supports the association between excessive VEGF expression and microaneurysm formation. Moreover, we reported the same effect in a prospective study with IV injection of ranibizumab in eyes with BRVO [109]. Furthermore, microaneurysms are seen frequently at the edge of the NPAs, which also indicates the association of VEGF with microaneurysm formation, because VEGF proteins are produced from the ischemic area.

Our previous data showed that older age was associated with microaneurysm formation, suggesting that older or weakened blood vessels are likely to form microaneurysms. Aging itself results in weakening of the function of pericytes [110].

OCTA also demonstrated an age-related reduction in retinal vascular density in normal human subjects [111]. While just a speculation, aging itself seems to be related to microaneurysm formation. Furthermore, previous animal studies have found that no microaneurysms were detected in an experimental RVO model, possibly because the animals had healthy vascular systems [112,113,114].

A histological study using postmortem diabetic eyes classified microaneurysms into three morphologic categories: saccular, fusiform, and focal bulge [115]. Additionally, Dubow et al. [104] morphologically classified microaneurysms from several retinal disorders into six types using adaptive optics scanning light ophthalmoscope FA: focal bulge, saccular, fusiform, mixed (saccular/fusiform), pedunculated, and irregular. Of all the microaneurysms in eyes with RVO, the focal bulge, saccular, and fusiform types occupied the majority, and the saccular type was the most prevalent [104]. This trend applied to subjects with other retinal disorders such as DR and hypertensive retinopathy [104].

Another morphologic classification of microaneurysms using OCTA was conducted in diabetic eyes [116]. The microaneurysms in eyes with BRVO also can be morphologically classified using OCTA images (Figure 18).

The VEGF levels in both the aqueous humor [3] and the vitreous cavity [118] were substantially higher in eyes with CRVO than in eyes with BRVO. In contrast, microaneurysm formation occurs more often in eyes with BRVO than eyes with CRVO. What is the cause of this? While this is only speculation, capillary endothelial cells, which regulate proliferation and angiogenesis due to upregulation of VEGF, are more severely and widely damaged in eyes with CRVO than in eyes with BRVO, and therefore, the endothelium perhaps cannot proliferate, although VEGF proteins are produced excessively.

Again, microaneurysms are often detected at the border between NPAs and the perfusion area, suggesting that VEGF proteins are produced from the NPAs, and the capillary endothelial cells in the neighboring area of perfusion may proliferate and cause microaneurysms and retinal NV. Increased concentrations of angiogenic lipid hydroperoxides may be present at the borders between the NPAs and perfused areas [119]. Even in eyes with CRVO, the eyes with nonischemic CRVO sometimes have a few microaneurysms, possibly because the endothelial cells in the perfused areas form microaneurysms in response to excessive VEGF proteins.

Microaneurysms in eyes with DR have been considered as early findings in several microvascular abnormalities. In contrast, microaneurysms in eyes with RVO are usually chronic, i.e., they are detected at around 6 months after disease onset [44]. Of course, this is because RVO is an acute disease, while diabetes mellitus is chronic. It usually takes several years for DR to develop after the onset of diabetes. Therefore, microaneurysm formation in diabetic eyes also requires a long time from the onset of diabetes. In eyes with RVO, microaneurysms were seen more frequently in the deep capillary plexus than in the superficial plexus [63], which is the same in diabetic eyes [120]. This is thought to occur because the densities of the capillaries are much greater in the deep capillary plexus than in the superficial capillary plexus [64]. The other horizontal location of microaneurysms, except at the edge of NPAs in eyes with RVO, is in the collateral vessels [55,63], which may differ from retinal disorders such as DR.

Retinal NV also develops due to overexpression of VEGF proteins similar to microaneurysm formation because VEGF plays a central role in angiogenesis. In addition, both retinal NV and microaneurysms form around the border between NPAs and perfused areas. In addition, retinal NV originates from the venous side [121] and microaneurysms commonly originate from the venous side of capillaries [55,56,100] (Figure 19).

Thus, both seem to have similar characteristics. The difference between retinal NV and microaneurysms is the location, i.e., retinal NV develops on the retina, while microaneurysms form in the retina, especially in the deep capillary plexus [63,120]. According to an analysis of eyes from diabetic patients, a histologic study [115] and an in vivo study using OCTA [120] also showed that microaneurysms tend to originate in the inner nuclear layer and its border zones, i.e., the deeper capillary plexus, which might implicate them in a hypoxic response [115] and consequent VEGF production. It has been suggested that microaneurysms may be aborted attempts at NV development [122].

Various modalities have been used to detect microaneurysms. Considering the treatment for ME due to leaky microaneurysms, it is obvious that the ability to determine which microaneurysms are at high risk for leakage and/or rupture is valuable. Fundus photographs are poor for detecting microaneurysms. FA is good for detecting leaky microaneurysms, especially in early-phase images, but fluorescein leakage can sometimes be an obstacle to detecting microaneurysms and provides incomplete information from the retinal deep capillary plexus [90]. OCT enables a differential layer analysis, but numerous B scan images are needed to detect microaneurysms. OCTA enables three-dimensional analysis, but the detection rate of microaneurysms is around 60% by FA. In contrast, ICGA not only provides information from the deep capillary plexus well, but it also detects leaky microaneurysms, and the effectiveness becomes more valuable when combined with the OCT color map [67,123,124] (Figure 6 and Figure 7). Early-phase FA images are generally used to detect microaneurysms. However, when those were compared with late-phase ICGA images, numerous hyperfluorescent spots were seen on the FA images but were limited on the ICGA images [125]. Moreover, the hyperfluorescent area on late-phase FA images did not always correspond to the lesions with ME on OCT, while hyperfluorescent spots on ICGA images may be responsible for ME [125]. Thus, both ICGA and OCT seem to be highly sensitive for determining which microaneurysms are responsible for ME. ICGA is more sensitive than fluorescein dye for detecting microaneurysms because the dye mostly binds to serum proteins such as albumin and lipoproteins [126]. Therefore, the dye hardly leaks through blood vessels, enabling ICGA to obtain sharp, detailed, and high-contrast images of aneurysms or leaky vessels [123] (Figure 6 and Figure 7). In leaky microaneurysms, albumin and lipoprotein carrying ICG are translocated from the blood vessels through permeable vessels walls and might be trapped in enveloping fibrin [125], which may account for the preferential detectability of leakage points responsible for ME [125].

Previous studies [44,127,128] have reported that microaneurysm formation is associated with persistent or refractory ME in eyes with BRVO. The presence of microaneurysms did not significantly affect the visual prognosis [44,127], which may be because ME, the factor that most affects the VA [32], was treated regardless of the presence of microaneurysms.

Microaneurysms have various types of morphologies and their sizes differ. The type of microaneurysm responsible for ME is controversial. Wang et al. [129] reported no correlation between leakage from microaneurysms and lumen diameter, wall diameter, or wall thickness, suggesting that the size may not be associated with the risk of leakage.

In contrast, Ezra et al. [130] suggested that the microaneurysm radius/associated vessel diameter ratio might be a useful metric for predicting leakage risk.

Alternatively, several neuroradiology studies have shown that multilobular aneurysms were more likely to rupture than unilobular aneurysms [131,132]. Another group determined that neck size but not the total size was more important to the risk of rupture [133]. Horii et al. [134] reported that microaneurysms with the ring sign (a high-intensity capsular structure on OCT images) in the eyes of diabetic patients were correlated positively with nearby retinal cystoid spaces. Hasegawa N et al. [120] showed that microaneurysms in the retinal deep capillary plexus are responsible for ME in the eyes of diabetic patients.

We recently investigated which type of microaneurysms were associated with macular edema in eyes with BRVO. Older age and the presence of microaneurysms in the collateral vessels were the independent predictive factors for retinal edema but not the size of microaneurysms or presence in the retinal deep capillary plexus [117].

### 3.9. Neovascularization

NV associated with RVO has two types: posterior segment NV and anterior segment NV. The former includes retinal NVD and retinal NV elsewhere (NVE); the latter NV includes NV of iris (NVI) and NV of angle (NVA). The development of both results from increased intraocular levels of VEGF, which is usually correlated with the area of retinal nonperfusion [13]. Therefore, nonischemic RVOs are not associated with either NV type, and macular BRVOs also are not even when the BRVO is ischemic because the area involved is small.

The type of NV that develops depends on the RVO subtype. Eyes with BRVO usually develop posterior segment NV and rarely develop anterior segment NV. Regarding posterior segment NV, NVE is more common than NVD in eyes with BRVO. In contrast, posterior segment NV after CRVO is more likely to occur at the optic disc than in the retina [135,136]. Eyes with CRVO usually develop NVD and anterior segment NV.

Generally, higher levels of VEGF are required to develop anterior segment NV than posterior segment NV [39]. Therefore, anterior segment NV often occurs in eyes with CRVO. Anterior segment NV includes NVA and NVI.

The frequency of detection of NVs depends on the modality used. Regarding the detection of posterior segment NVs, slit-lamp biomicroscopy is the least sensitive and FA is more sensitive because retinal NV associated with RVO lacks tight junctions [137], resulting in marked hyperfluorescence (leakage) from the NVs (Figure 20), which differs from collateral vessels. OCTA also can detect posterior segment NVs. Sogawa et al. [121] reported the effectiveness of OCTA in visualizing NVE in an eye with hemi-CRVO. The advantages of OCTA are that the NV network is visualized clearly because of lack of fluorescein dye and B scan images demonstrate that the NV is on the surface of the retina or optic disc, which is also effective for distinguishing NVs from microaneurysms or collateral vessels.

Regarding anterior segment NVs, slit-lamp biomicroscopy is the least sensitive, iris and angle FA are more sensitive, and histopathological examination of surgical specimens or postmortem tissues is the most sensitive [138]. OCTA has been used to visualize iris vessels in different ocular disorders from RVOs [139,140], which may become the other option for detecting iris NVs.

Regarding posterior segment NV, 24% to 40% of eyes with BRVO [141,142,143] and 10% to 26% of eyes with CRVO [39,53,135,136,143,144,145,146,147] develop posterior segment NV.

NVD or NVE development may depend on the proximity to the occlusion site. Of eyes that developed posterior segment NV associated with BRVO, 71% developed NVE alone, 19% NVE and NVD together, and 10% NVD alone [142]. Of 144 eyes with untreated CRVO, 24% developed NVD and 2% NVE [148]. The relative infrequency of posterior segment NV compared to anterior segment NV in cases of ischemic CRVO has been attributed to the relative absence of viable retinal capillary endothelial cells after ischemic CRVO [135], which seems to be the same as the pathophysiology of microaneurysm formation.

The proportion of eyes with posterior segment NV in RVO cases depends on the status of the vitreous [38,39,149]. In eyes with BRVO, Avunduk et al. [38] reported that the proportion of posterior segment NV in cases with a complete posterior vitreous detachment (PVD) was significantly lower (17%) than in cases with a partial PVD (58%). In eyes with CRVO, posterior segment NV developed only in cases with ischemic CRVO in which the PVD was incomplete [39]. No patients with a complete PVD developed posterior segment NV compared to 57% of the cases with a partial PVD, which was a significant difference [39]. Thus, the vitreous scaffold is necessary for development of posterior segment NV.

Sixty to ninety percent of eyes with BRVO in which NVE and/or NVD is untreated develop vitreous hemorrhage (VH) [150,151]. Besides, it was reported that about 7% of cases of BRVO developed a VH, and in most cases from posterior segment NV [152]. We previously reported microvascular abnormalities in eyes with major BRVO and the ischemic type using ultrawide field FA (UWFFA) (Figure 20). The study showed that seven of 40 eyes (18%) developed posterior NV and five of 40 eyes (13% of all cases and 71% of the cases with NV) developed a VH [153]. All posterior NVs were NVE and the NVEs were in Zone 1 or 2. In other words, NVs were undetected in the peripheral retina (Zone 3), and microaneurysms also were detected significantly less in the peripheral retina than in the posterior pole. Thus, the peripheral retina already appeared to be non-functioning (Figure 20). In addition, all cases with a VH had NPAs exceeding 5 disc diameters in Zone 1 [153], suggesting that posterior ischemia but not peripheral ischemia was associated with development of VH.

**Figure 20 jcm-10-00405-f020:**
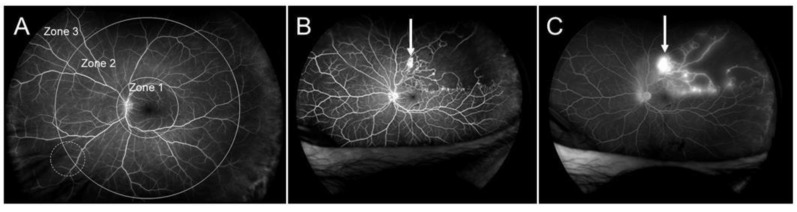
Zone grading and ultrawide field fluorescein angiography (UWFFA) findings in eyes with branch retinal vein occlusion. (**A**) Zone grading on UWFFA. Zone 1 has a radius of 3 disc diameters and roughly corresponds to the posterior pole. Zone 2 extends from the edge of Zone 1 anteriorly with a radius of 9 disc diameters and overlaps the vortex veins (white dotted circle). Zone 3 is the region anterior to Zone 2, as previously reported [153]. (**B**) Early phase of fluorescein angiogram. (**C**) Late phase of a fluorescein angiogram. White arrows indicate retinal neovascularization. Retinal nonperfused areas are widely spread, but microaneurysms or neovascularization are not seen in Zone 3. Reproduced from Yasuda et al. [153] with the permission of the publisher.

Development of anterior segment NV depends on the VEGF concentration in the aqueous humor [154]. Retinal ischemia and decreased blood flow in eyes with RVO induce production of VEGF in the retina, which diffuses into the vitreous gel and, ultimately, the aqueous humor. It was reported that VEGF concentration increased in the vitreous fluid in ischemic RVO, while the concentration in the aqueous humor was about 60% lower than that in the vitreous cavity [108]. Furthermore, for anterior segment NV to develop, the VEGF level should be elevated and continuously present to persist [155]. Therefore, it is reasonable that anterior segment NV is more common after CRVO than BRVO, presumably reflecting the higher levels of intraocular VEGF in eyes with CRVO than in eyes with BRVO [39]. Generally, anterior segment NV does not develop after macular BRVO, rarely develops after major BRVO, uncommonly after hemi-CRVO, and commonly after CRVO. Thus, anterior segment NV is primarily in eyes with CRVO, but this is not the case in eyes with nonischemic CRVO. It has been reported that anterior segment NV develops in 20% to 70% of cases of ischemic CRVO [60,136]. Anterior segment NV occurs at the pupillary margin in 44%, in the angle in 12%, and in both in 44% of cases, and NVA can develop before NVI [156].

Collateral vessels occur more often at the optic disc in eyes with CRVO than in eyes with BRVO. The presence of collateral vessels at the optic disc in eyes with CRVO is a negative predictor of development of anterior segment NV, suggesting that optic disc collaterals may protect against ischemia [92].

The development of posterior segment NV in RVO does not depend on the status of the vitreous [39], which might be a matter of course because anterior segment NV does not need a vitreous scaffold. The presence of anterior segment NV is associated with a high risk of developing neovascular glaucoma (NVG), a severe complication of ischemic retinal disorders. NVG developed in 23% to 60% of cases with ischemic CRVO [157] (McIntosh et al., 2010). Hayreh et al. [143,158] documented the time course of development of each NV; the incidence of anterior segment NV is the highest within the first 100 days and remains through 6 months [159].

## 4. Follow-Up Using Multiple Modalities

Anti-VEGF therapy provides substantial visual improvements in eyes with ME associated with RVO. Therefore, it is important to maintain or enhance the efficacy during follow-up. ME is most closely associated with the VA in eyes with RVO and is most effectively quantified by OCT. OCT measurement is rapid and non-invasive and so frequent examination is possible. The CST measured by OCT or exudation on an OCT color map is useful to determine the need for retreatment. Reduced follow-up and fewer injections of anti-VEGF drugs were associated with decreased vision in CRVO [160,161]. In the Rubeosis Anti-VEgf (RAVE) trial, when ranibizumab injections were withheld (months 9–11), about half of the patients had recurrent ME with subsequent loss of the initial VA gains. Once ranibizumab therapy was restarted, anatomic and VA improvements were restored in most eyes [161]. Therefore, frequent examinations and repeated injections are needed.

FA and ICGA should be performed early after the initial visit if patients do not have a drug allergy, liver and/or renal functional disorder, and are not pregnant. Angiograms are useful for clinicians to distinguish RVO from other diseases, define the subtype (BRVO, CRVO, or hemi-CRVO, and ischemic or nonischemic), and determine the presence of NVs and leakage points causing ME. Thus, because angiograms can provide valuable information, frequent examinations are desirable. However, FA requires intravenous dye injection, which rarely causes serious adverse side effects such as allergy, cutaneous rashes, Quincke edema, infarctions, pulmonary edema, or anaphylactic shock, and mild adverse effects such as nausea and hives are common [63,162]. Therefore, clinicians usually hesitate to perform FA frequently due to the invasive nature. In contrast, frequent examination of OCTA is possible because of the non-invasive nature. In fact, OCTA is advantageous because it detects longitudinal vascular changes in eyes with RVO [44,55,69,163] (Figure 21). The disadvantages of OCTA are the limited field of view and inability to detect dynamic changes such as leakage [63]. However, the field of view is expanding (Figure 22) and the efficacy of widefield OCTA has been increasingly reported [164,165,166]. Furthermore, the dynamic changes are evaluable by supplement of the other modalities.

When managing eyes with CRVO, it is important to judge the degree of retinal ischemia and prevent NVG progression. Clinicians should carefully manage patients with CRVO during the early stage, particularly within 6 months, because that is when severe complications occur [158]. Larger blot hemorrhages were associated with retinal nonperfusion in eyes with CRVO and can be an indicator of retinal ischemia [167]. Furthermore, OCTA may show that the retinal nonperfusion expanded in eye with CRVO, probably indicating conversion from nonischemic to the ischemic form [168].

FA and/or ICGA, which cannot be performed frequently, are useful for detecting focal leakage points in eyes with RVO, which may help in choosing the treatment modalities.

## 5. Conclusions and Future Directions

Anti-VEGF therapy, which provides rapid resolution of ME and much better visual improvements than conventional treatments in eyes with RVO, is now a first-line therapy.

However, several problems related to anti-VEGF therapy have been raised, e.g., high cost, high disease recurrence rate, need of frequent visits and injections, concerns regarding systemic absorption, non-responders, and tachyphylaxis. To address these, cost reductions, development of long-acting drugs and a sustained-release drug delivery system, and identification of new targets for the therapy other than VEGF are required.

Alternative therapies are needed for patients who are non-responders or develop tachyphylaxis to the anti-VEGF drug. We previously investigated the relation between chronic ME and microvascular abnormalities in eyes with BRVO using OCT, OCTA, FA, and ICGA and reported that microaneurysm formation was associated with refractory ME [44], which was consistent with other reports [127,128]. It is important to suppress microaneurysm formation by prompt initiation of anti-VEGF agents [44]. Direct laser photocoagulation targeting leaky microaneurysms decreases the need for additional treatment for ME [44].

Development of alternative new therapies also may be ideal for avoiding these problems, for which elucidation of the pathology of RVO is essential. Ocular imaging technologies have progressed markedly. A multimodal imaging technique using various modalities and vitreous or aqueous and postmortem samples from patients can be more useful for elucidating the pathology and will likely help lead to development of new therapies.

The current therapeutic strategy in eyes with RVO is reduction of the ME. Apart from this, is improvement of microcirculatory disorder or guidance of new vessels into the ischemic area possible? Although macular ischemia, atrophy, or photoreceptor degeneration other than ME can decrease visual function, additional therapies to preserve photoreceptor function should also be developed.

Future directions may include ocular imaging technologies, application of artificial intelligence in disease management, and identification of new targets other than VEGF. If we can determine individual patient reactivity to drugs, individualized medical care may be possible.

## Figures and Tables

**Figure 1 jcm-10-00405-f001:**
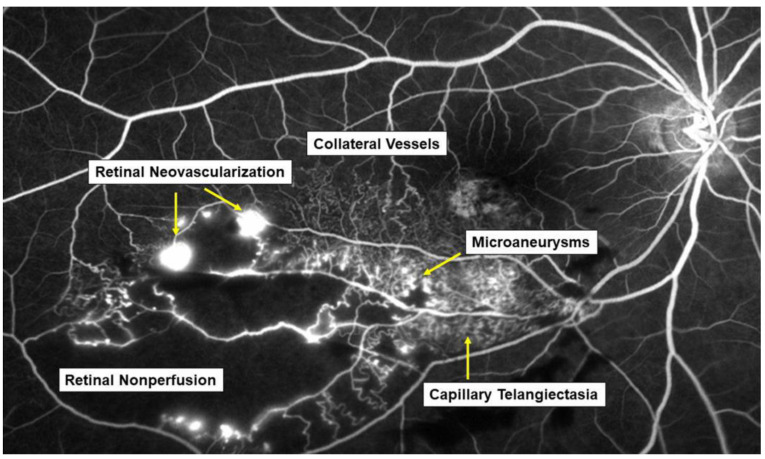
Various microvascular abnormalities associated with branch retinal vein occlusion (BRVO) on a fluorescein angiogram. In the chronic stage, various microvascular abnormalities are observed in an eye with BRVO.

**Figure 2 jcm-10-00405-f002:**
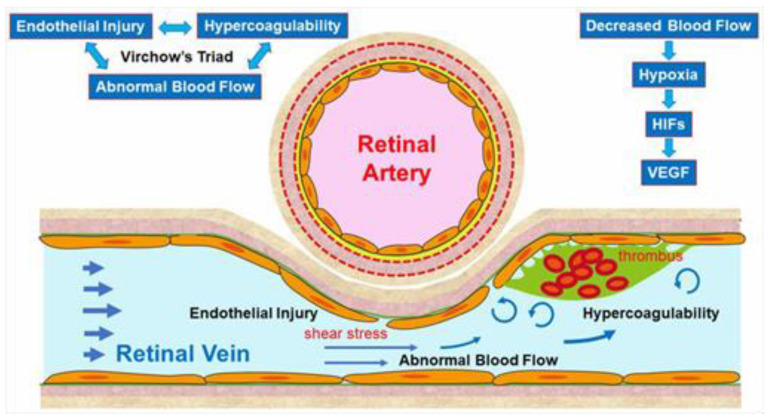
Mechanism of retinal vein occlusion (RVO). A rigid artery compresses the underlying venous wall at the arteriovenous crossings, resulting in disturbing the venous return and changing the course of the venous flow, turbulent blood flow, causing chronic damage to the retinal vascular endothelial cells, and thrombosis formation, which result in the onset of RVO. A thrombosis is thought to be caused by the three important factors of Virchow’s triad, i.e., endothelial injury, hypercoagulability, and abnormal blood flow. HIF: hypoxia inducible factor, VEGF: vascular endothelial growth factor.

**Figure 3 jcm-10-00405-f003:**
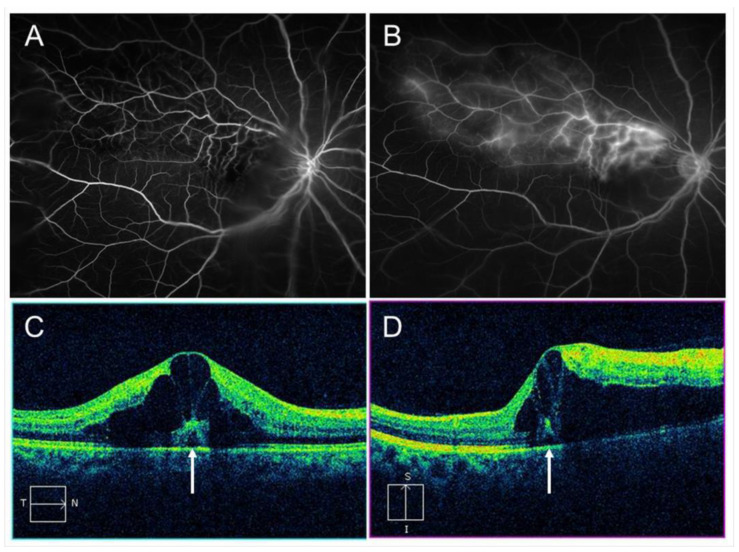
Macular edema (ME) associated with branch retinal vein occlusion. (**A**) Early phase of fluorescein angiogram. (**B**) Late phase of fluorescein angiogram. The hyperfluorescence on (**B**) indicates pooling due to hyperpermeability from retinal vessels. (**C**) Horizontal scan of optical coherence tomography (OCT) image. (**D**) Vertical scan of OCT image. Cystoid macular edema and tiny serous retinal detachment (white arrows) are observed.

**Figure 4 jcm-10-00405-f004:**
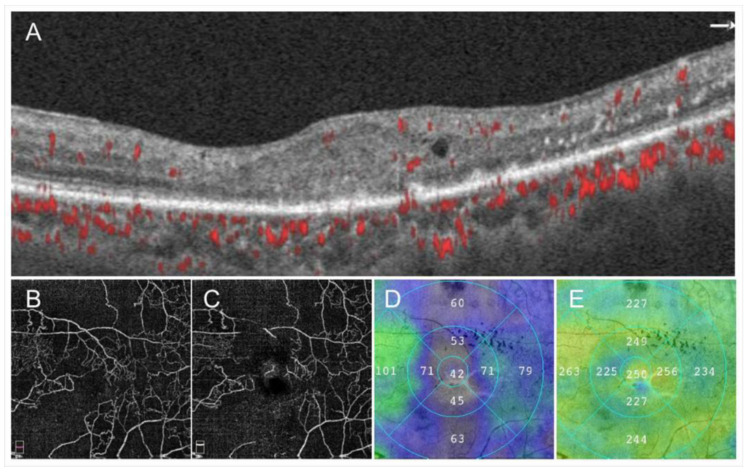
Images from a patient with ischemic central retinal vein occlusion. (**A**) B Scan of optical coherence tomography (OCT) angiogram. Red dots indicate the presence of flow signals. Each retinal structure at the macula is disrupted. Especially the injury of outer retina is remarkable. Macular edema is not evident that much. (**B**) OCT angiogram in the superficial capillary plexus. (**C**) OCT angiogram in the deep capillary plexus. Widespread non-perfused areas are observed. (**D**) OCT color map from the internal limiting membrane to the inner plexiform layer. (**E**) OCT color map of the whole retina. The inner retina remarkably became thin (**D**). The best-corrected visual acuity was 20/200.

**Figure 5 jcm-10-00405-f005:**
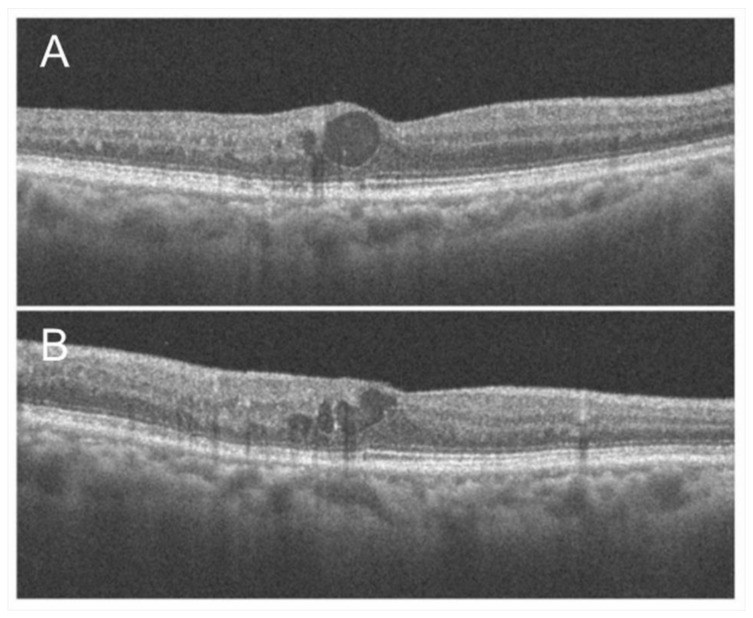
Optical coherence tomography findings of chronic macular edema in an eye with branch retinal vein occlusion. (**A**) Horizontal scan. (**B**) Vertical scan. Although cystoid macula edema is observed, the foveal photoreceptors are relatively intact, and the best-corrected visual acuity was 20/16.

**Figure 6 jcm-10-00405-f006:**
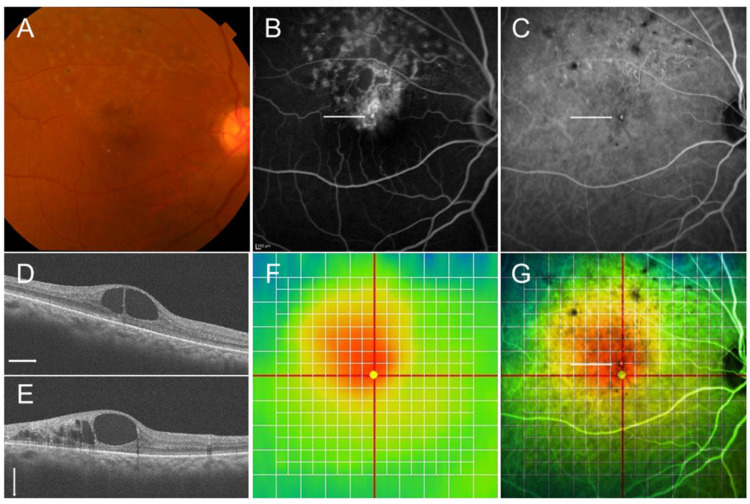
Multimodal imaging in an eye with chronic macular edema after branch retinal vein occlusion. (**A**) Color fundus photograph. (**B**) Late-phase fluorescein angiogram. (**C**) Late-phase indocyanine green angiogram. Hyperfluorescent dot (white arrows on (**B**,**C**)) indicates leaky microaneurysm. (**D**) Optical coherence tomography (OCT) horizontal scan. (**E**) OCT vertical scan. (**F**) OCT color map. (**G**) Overlaid image with (**C**,**F**). The white dot (white arrow) was located around the center of macular edema.

**Figure 7 jcm-10-00405-f007:**
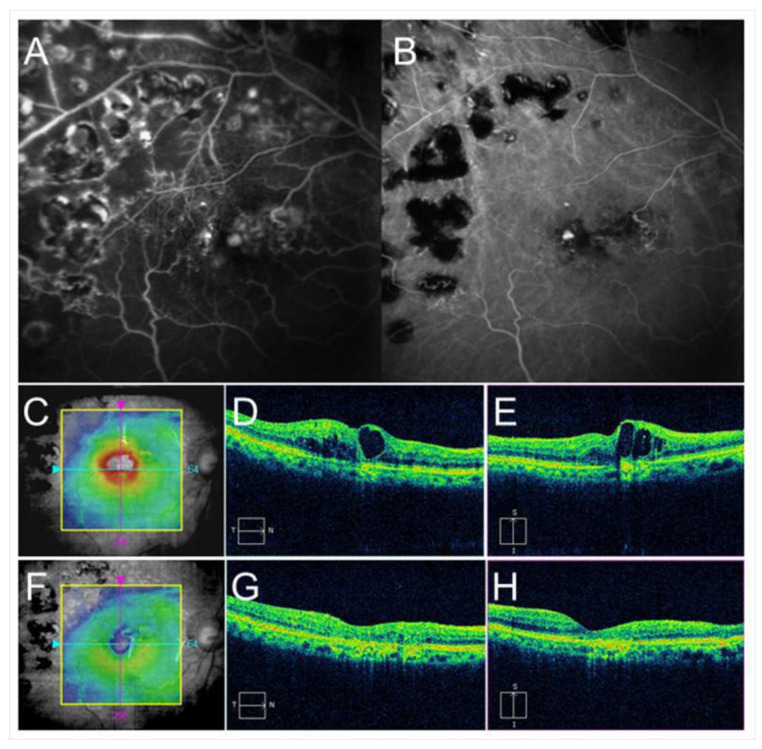
Before and after targeting laser in an eye with chronic macular edema after branch retinal vein occlusion. (**A**) Late-phase fluorescein angiogram. (**B**) Late-phase indocyanine green angiogram (ICGA). Hyperfluorescent spots are evident on ICGA (**B**). (**C**–**E**) Before targeting laser. (**F**–**H**) After targeting laser. (**C**,**F**) Optical coherence tomography (OCT) color maps. (**D**,**G**) OCT horizontal scans. (**E**,**H**) OCT vertical scans. Macular edema was almost resolved after targeting laser (**F**–**H**).

**Figure 8 jcm-10-00405-f008:**
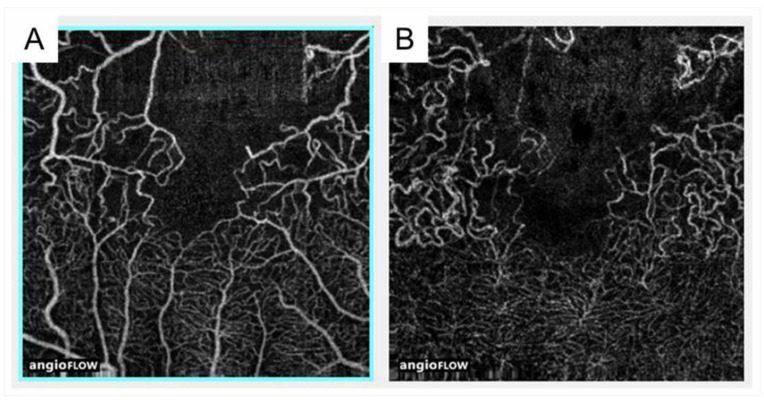
Optical coherence tomography angiography findings in an eye with branch retinal vein occlusion. (**A**) Optical coherence tomography (OCT) angiogram in the superficial capillary plexus. (**B**) OCT angiogram in the deep capillary plexus. Foveal capillary ring was disrupted and the border between foveal avascular zone and nonperfused area was not identified. Capillary telangiectasia was remarkably widely observed in the deep capillary plexus.

**Figure 9 jcm-10-00405-f009:**
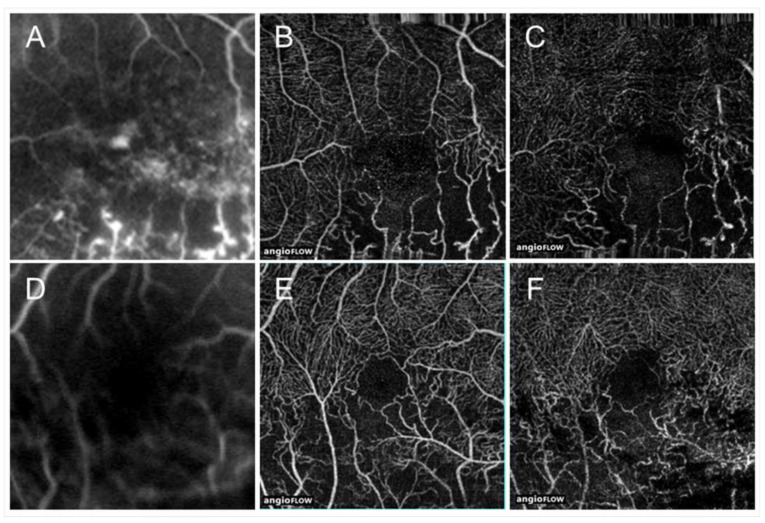
Comparison of optical coherence tomography angiography with fluorescein angiography in eyes with branch retinal vein occlusion. (**A**,**D**) Fluorescein angiograms. (**B**,**E**) Optical coherence tomography (OCT) angiograms in the superficial capillary plexus. (**C**,**F**) OCT angiograms in the deep capillary plexus. Nonperfused areas and foveal avascular zones were clearly visualized on the OCT angiograms. Reproduced from Suzuki et al. [63] with the permission of the publisher.

**Figure 10 jcm-10-00405-f010:**
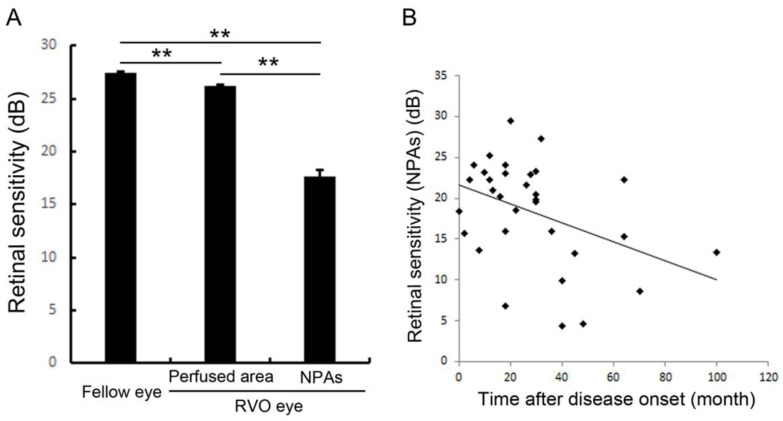
Retinal sensitivities in the nonperfused area and comparison with fellow eyes and perfused area. (**A**) The mean retinal sensitivity in the fellow eyes and eyes with retinal vein occlusion (RVO). Error bars represent the standard errors of the mean above. The mean retinal sensitivity levels in eyes with RVO, even in the perfused areas, were significantly (*p* < 0.0001 **) lower than in the fellow eyes. (**B**) The relationship between mean retinal sensitivity in nonperfused areas in eyes with RVO and the duration after the disease onset. There was an inverse correlation (r = 0.379, *p* = 0.0391 *) between the retinal sensitivity and the time after disease onset. dB: decibel; NPA: nonperfused area; RVO: retinal vein occlusion. Reproduced from Tomiyasu et al. [67] with the permission of the publisher.

**Figure 11 jcm-10-00405-f011:**
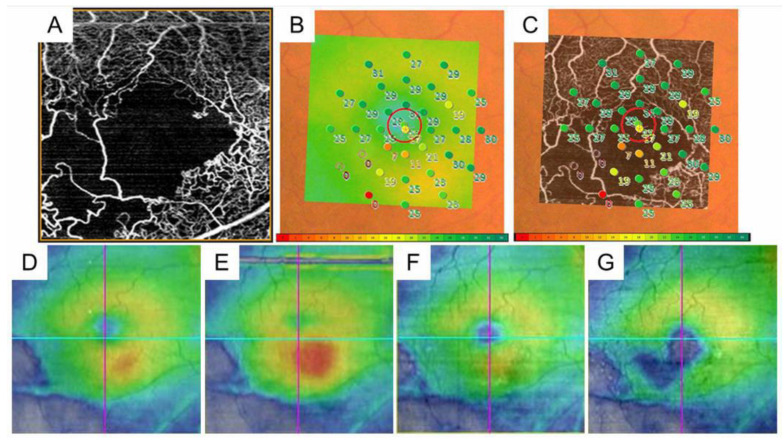
The retinal sensitivities in the nonperfused areas and the relationship with macular edema in eyes with branch retinal vein occlusion. (**A**) Optical coherence tomography (OCT) angiogram (3 × 3 mm) centered on the fovea. (**B**) Fundus-monitoring microperimetry (MP-3, Nidek, Gamagori, Japan) image evaluating 33 stimulus points covering the central 5 °. (**C**). A superimposed image of the MP-3 and OCT angiogram. (**D**) OCT color map obtained at the same time as (**A**–**C**). (**E**) Three months after treatment. (**F**) Twelve months after treatment. (**G**) Twenty-four months after treatment. The area where the retinal sensitivities decreased to zero has never had recurrent macular edema and gradually becomes thin.

**Figure 12 jcm-10-00405-f012:**
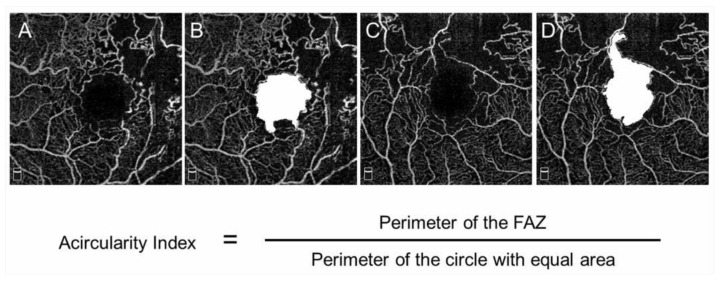
Acircularity of foveal avascular zone in eyes with branch retinal vein occlusion. (**A**–**D**) Optical coherence tomography (OCT) angiogram in the superficial layer. The foveal avascular zones (FAZs) are painted in white (**B**,**D**). The size of the FAZ area is 0.62 (**B**) and 0.84 mm^2^ (**D**), respectively. The acircularity index is 1.09 (**B**) and 1.50 (**D**), respectively.

**Figure 13 jcm-10-00405-f013:**
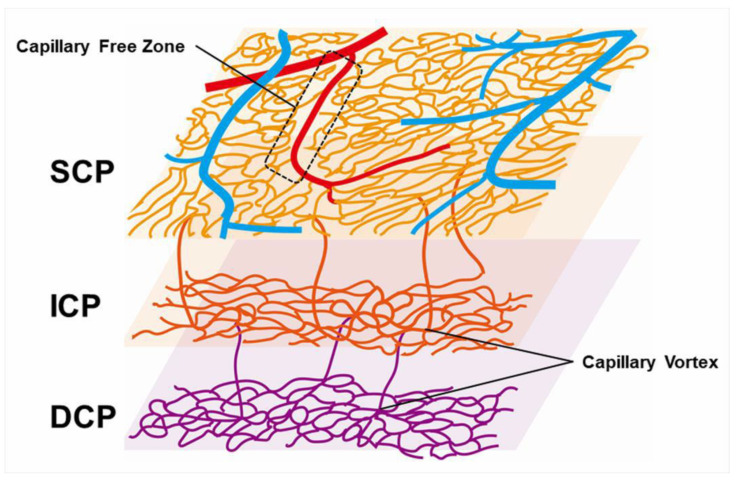
Schematic representation of retinal vascular structure. Retinal arteries are distinguished from veins because the capillary free zone is located around the artery. SCP: superficial capillary plexus; ICP: intermediate capillary plexus; DCP: deep capillary plexus.

**Figure 14 jcm-10-00405-f014:**
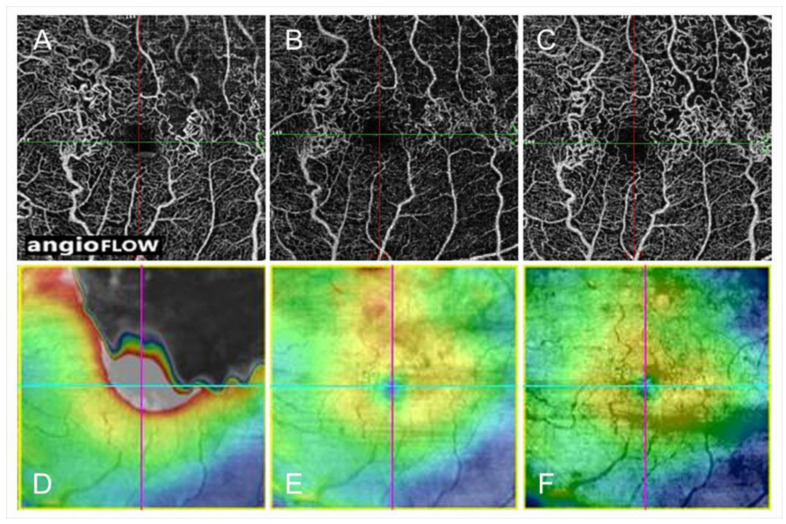
Optical coherence tomography angiography (OCTA) and OCT color map findings in eyes with collateral vessels after branch retinal vein occlusion. (**A**–**C**) OCTA images. (**D**–**F**) OCT color maps. (**A**,**D**) Baseline. (**B**,**E**) Month 3. (**C**,**F**) Month 12. Collateral vessels are seen in both sides of the foveal avascular zone. The collateral vessels are present at baseline and gradually matured over time.

**Figure 15 jcm-10-00405-f015:**
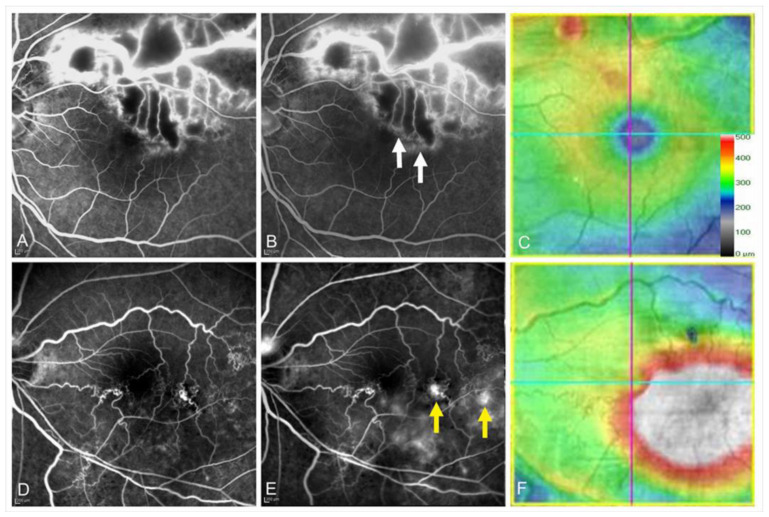
Leakage from collateral vessels in eyes with branch retinal vein occlusion. (**A**–**C**) Images from an eye with non-leaky collateral vessels. (**D**–**F**) Images from an eye with leaky collateral vessels. (**A**,**D**) Early phase of fluorescein angiograms. (**B**,**E**) Late phase of fluorescein angiograms. (**C**,**F**) Optical coherence tomography (OCT) color maps. Both eyes had microaneurysms in the collateral vessels. The collaterals were not leaky (**A**–**C**), whereas the collateral vessels were leaky (**D**–**F**). White arrows on (**B**) and yellow arrows on (**E**) indicate non-leaky and leaky microaneurysms, respectively. Reproduced from Suzuki et al. [55] with the permission of the publisher.

**Figure 16 jcm-10-00405-f016:**
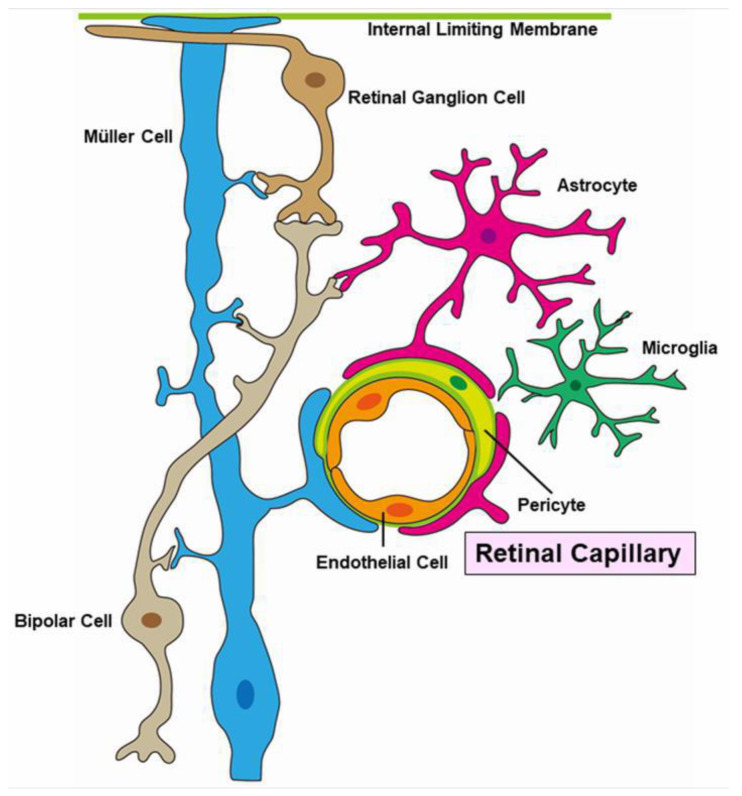
Schematic representation of a neuro-vascular unit.

**Figure 17 jcm-10-00405-f017:**
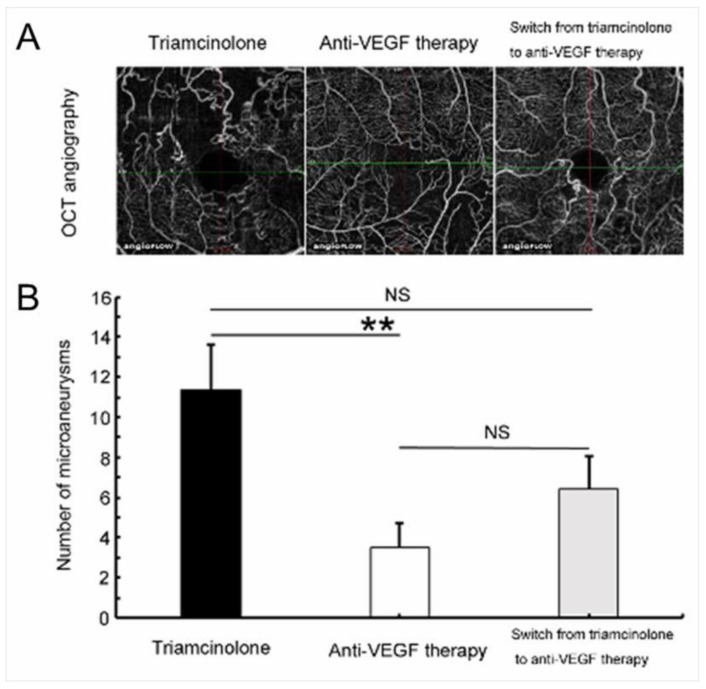
Anti-vascular endothelial growth factor (VEGF) therapy reduced microaneurysm formation in eyes with branch retinal vein occlusion. (**A**) Representative optical coherence tomography (OCT) angiograms from each treatment group. (**B**) The number of microaneurysms after each treatment. Anti-VEGF therapy significantly (** *p* < 0.01) reduced microaneurysm formation compared with the Triamcinolone group. Error bars represent the standard errors of the mean above. NS: not significant. Reproduced from Tomiyasu et al. [44] with the permission of the publisher.

**Figure 18 jcm-10-00405-f018:**
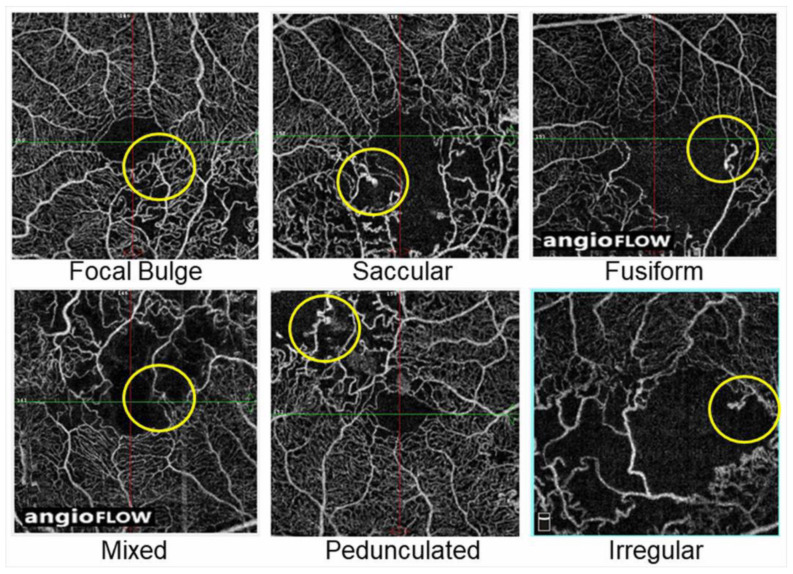
Representative morphological patterns of microaneurysms on optical coherence tomography angiograms in eyes with branch retinal vein occlusion. Reproduced from Esaki et al. [117] with the permission of the publisher.

**Figure 19 jcm-10-00405-f019:**
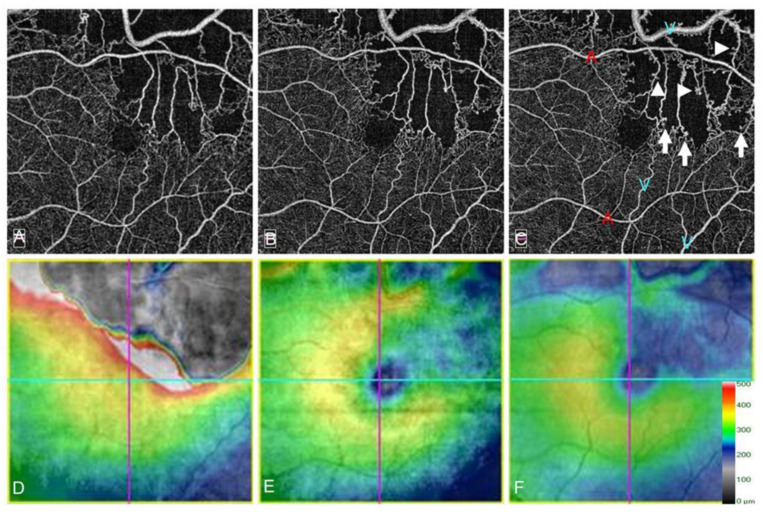
Microaneurysm formation on optical coherence tomography angiograms (OCTA) in an eye with branch retinal vein occlusion. (**A**–**C**) OCTA images. (**D**–**F**) OCT color maps. (**A**,**D**) Baseline. (**B**,**E**) Month 6. (**C**,**F**) Month 12. Microaneurysm formation could be observed on longitudinal OCTA images. Some microaneurysms were in the collateral vessels (**A**–**C**) and the others at the edge of nonperfused areas (arrowheads on (**C**)). Red-colored “**A**”s and blue-colored “**V**”s on **C** indicate the arterioles and venules, respectively. All microaneurysms originated from the venous walls (**C**). Reproduced from Suzuki et al. [55] with the permission of the publisher.

**Figure 21 jcm-10-00405-f021:**
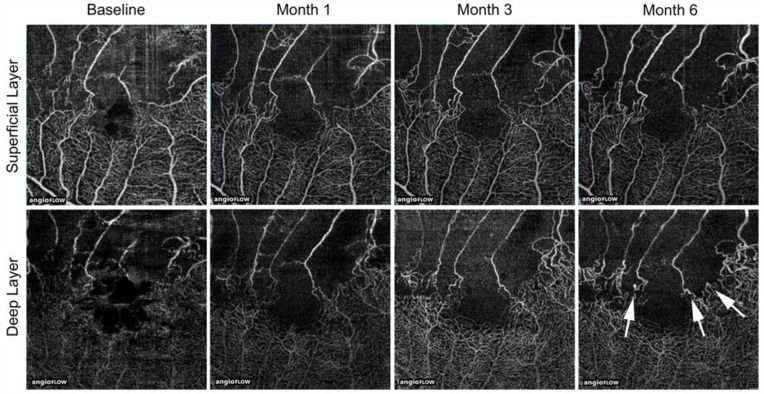
Longitudinal changes of microvascular abnormalities seen on optical coherence tomography (OCT) angiograms over time in eyes with branch retinal vein occlusion. Microaneurysms (white arrows) were seen in the deep capillary plexus at month 6. Reproduced from Tomiyasu et al. [44] with the permission of the publisher.

**Figure 22 jcm-10-00405-f022:**
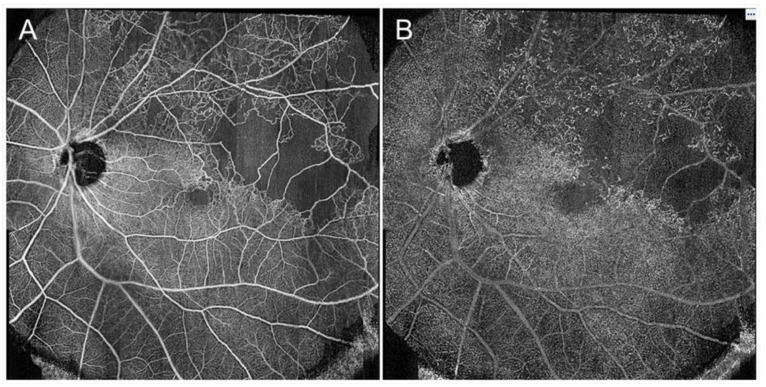
Widefield optical coherence tomography (OCT) angiography findings in eyes with branch retinal vein occlusion. (**A**) Retinal superficial capillary plexus. (**B**) Retinal deep capillary plexus. The images were obtained using PLEX^®^ Elite (Carl Zeiss Meditec, Jena, Germany) and montaged from two pieces of 15 × 9 mm images.

## Data Availability

No new data were created or analyzed in this study. Data sharing is not applicable to this article.

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
