# Peer review of "Multimodal Imaging of Microvascular Abnormalities in Retinal Vein Occlusion"

_jcm, 2021, doi:10.3390/jcm10030405_

Round 1

Reviewer 1 Report

The paper is well written and provides valuable information regarding the diagnosis and treatment of retinal vein occlusion. I have a few comments.

Line 33 Please add the reference regarding the non-responders of anti-VEGF treatment for RVO.

Line 78 This sentence does not make sense. What is vascular cells ? ECs, pericytes, and glia consisting the blood vessels in retina? or just ECs ? injured EC may play a role in abnormal microcirculation.

Line 156-162  The author described that VEGF stimulates VEGFR2 and NO production. The author may well add the sentence that VEGF also increase the expression of ICAM-1.

Line 790 Please add the references.

Author Response

Dear Reviewer 1,

We appreciate the comments of the reviewer. Specific concerns of the reviewer are addressed below:

  1. Line 33 Please add the reference regarding the non-responders of anti-VEGF treatment for RVO.

              We really thank the reviewer for the significant comments. Since there are no valuable the references regarding the non-responders of anti-VEGF treatment for RVO, we delete the sentence.

  1. Line 78 This sentence does not make sense. What is vascular cells ? ECs, pericytes, and glia consisting the blood vessels in retina? or just ECs ? injured EC may play a role in abnormal microcirculation.

              We really thank the reviewer for the significant comments. It means ECs. We revise the terminology.

  1. Line 156-162  The author described that VEGF stimulates VEGFR2 and NO production. The author may well add the sentence that VEGF also increase the expression of ICAM-1.

              We really thank the reviewer for the significant comments. We agree with the reviewer and add the sentence as the reviewer suggested.

  1. Line 790 Please add the references.

              We really thank the reviewer for the significant comments. We add some references and change the sentences.

Reviewer 2 Report

Jcm-1066760

Thank you for giving me the opportunity of reviewing this article. This article is well written. The authors summarize all common findings in both BRVO and CRVO with multimodal imaging techniques. Furthermore, they describe the pathological point of view. This review article may help a comprehensive understanding of retinal vein occlusion. I want to ask a couple of questions that may help to improve this article.

In line 74 and 137, the authors stated that venous narrowing is the major factor in the pathogenesis of BRVO. I agree the venous narrowing is one of the important factors that lead to BRVO. However, the artery overcrossing type does not have severe venous narrowing, unlike the venous overcrossing type, which suggests that the other factors, such as increasing share stress due to abnormal course of vessel and endothelial cell damage, might cause the pathogenesis of BRVO. The pathogenesis of BRVO might still be controversial; thus, the authors should add these possibilities to discuss the pathogenesis of BRVO.

In paragraph 2.4., the authors demonstrated that the abnormal blood flow was evaluated with several instruments. It is recommended that the authors give more information about each result, especially about which vessel they measured.

In paragraph 3.5., the authors should describe the limitations of FAZ that large variation between individuals and only one anatomical FAZ. Although many studies that ignore these limitations have been reported, readers have to consider these limitations.

In paragraph 3.6. and 3.7., authors well summarize the capillary telangiectasia and collateral vessels in human and animal subjects. I generally agree with the author’s description. One question in these paragraphs is that the difference between the capillary telangiectasia and the collateral vessels. To the best of my knowledge, there is no distinct difference between the two abnormal vessels. My thought is that the capillary telangiectasia or dilated capillary have greater caliber capillaries, which might be over 15 microns because the normal capillary caliber in humans is 5-10 microns. In contrast, the collateral vessel could be observed with ophthalmoscopy or fundus photo, which means that the vessel caliber might be almost the same as retinal venules or more. If authors have a distinct definition, I would like to know it. Otherwise, the duration of collateral formation or frequency of collateral vessel is unclear.

Author Response

Dear Reviewer 2,

We appreciate the comments of the reviewer. Specific concerns of the reviewer are addressed below:

  1. In line 74 and 137, the authors stated that venous narrowing is the major factor in the pathogenesis of BRVO. I agree the venous narrowing is one of the important factors that lead to BRVO. However, the artery overcrossing type does not have severe venous narrowing, unlike the venous overcrossing type, which suggests that the other factors, such as increasing share stress due to abnormal course of vessel and endothelial cell damage, might cause the pathogenesis of BRVO. The pathogenesis of BRVO might still be controversial; thus, the authors should add these possibilities to discuss the pathogenesis of BRVO.

              We really agree with the reviewer’s significant comments. We revise these sentences regarding possibilities to discuss the pathogenesis.

  1. In paragraph 2.4., the authors demonstrated that the abnormal blood flow was evaluated with several instruments. It is recommended that the authors give more information about each result, especially about which vessel they measured.

              We really thank the reviewer for the significant comments. We add the information about which vessels were used for measurement of the blood flow using several instruments.

  1. In paragraph 3.5., the authors should describe the limitations of FAZ that large variation between individuals and only one anatomical FAZ. Although many studies that ignore these limitations have been reported, readers have to consider these limitations.

              We thank the reviewer for the significant comments. We agree with the reviewer’s suggestion and address the limitations regarding FAZ measurement. .

  1. In paragraph 3.6. and 3.7., authors well summarize the capillary telangiectasia and collateral vessels in human and animal subjects. I generally agree with the author’s description. One question in these paragraphs is that the difference between the capillary telangiectasia and the collateral vessels. To the best of my knowledge, there is no distinct difference between the two abnormal vessels. My thought is that the capillary telangiectasia or dilated capillary have greater caliber capillaries, which might be over 15 microns because the normal capillary caliber in humans is 5-10 microns. In contrast, the collateral vessel could be observed with ophthalmoscopy or fundus photo, which means that the vessel caliber might be almost the same as retinal venules or more. If authors have a distinct definition, I would like to know it. Otherwise, the duration of collateral formation or frequency of collateral vessel is unclear.

              We really thank the reviewer for the significant comments. Actually, it is very difficult to distinguish collateral vessels from capillary telangiectasia as the reviewer pointed out. We add the definition of collateral vessels in the paragraph 3.7.

Reviewer 3 Report

Dear Editor

This is an interesting review. However, I have some minor issues to be solved:

- I suggest the authors to add these papers in the pathogenesis of retinal vein occlusions and discuss them (doi: 10.1097/IAE.0000000000000584, doi: 10.1016/j.ajo.2015.04.019, doi: 10.1167/iovs.17-22229 and doi: 10.1155/2019/3982428)

- Page 5 line 130 “3. Multimodal imaging of microvascular abnormalities in retinal vein occlusion” and the next two sections are: 3.1 Thrombosis and 3.2. Macular Edema. I am not fully convinced that this is the appropriate position for these two sections.

- Figure 6, the arrow indicates the leaking microaneurysm on ICGA. However, it is not leaking on ICGA, I suggest to replace the arrow in the FA image.

- About OCTA I suggest the authors to add the discussion of the widefield OCTA and these papers should appropriately comment (doi: 10.1097/IAE.0000000000002993, doi: 10.1371/journal.pone.0214892 and doi: 10.1038/s41433-020-01317-9)

Author Response

Dear Reviewer: 3,

We appreciate the comments of the reviewer. Specific concerns of the reviewer are addressed below:

  1. - I suggest the authors to add these papers in the pathogenesis of retinal vein occlusions and discuss them (doi: 10.1097/IAE.0000000000000584, doi: 10.1016/j.ajo.2015.04.019, doi: 10.1167/iovs.17-22229 and doi: 10.1155/2019/3982428)

              We really thank the reviewer for the significant suggestion. We add these papers in the pathogenesis and discuss about them.

  1. - Page 5 line 130 “3. Multimodal imaging of microvascular abnormalities in retinal vein occlusion” and the next two sections are: 3.1 Thrombosis and 3.2. Macular Edema. I am not fully convinced that this is the appropriate position for these two sections.

              We thank the reviewer for the significant comments. We really agree with the reviewer. We change the position of these two sections.

  1. - Figure 6, the arrow indicates the leaking microaneurysm on ICGA. However, it is not leaking on ICGA, I suggest to replace the arrow in the FA image.

              We thank the reviewer for the significant comment. We really agree with the reviewer. We revise the figure 6 on which the arrows indicating leaky microaneurysm are placed in both the FA and ICGA images.

  1. - About OCTA I suggest the authors to add the discussion of the widefield OCTA and these papers should appropriately comment (doi: 10.1097/IAE.0000000000002993, doi: 10.1371/journal.pone.0214892 and doi: 10.1038/s41433-020-01317-9)

              We really thank the reviewer for the significant comment. We really agree with the reviewer. We add these papers and discuss about them.